# Machine Learning Approach to Real-Time 3D Path Planning for Autonomous Navigation of Unmanned Aerial Vehicle

**Abera Tullu [1], Bedada Endale [2], Assefinew Wondosen [2] and Ho-Yon Hwang [3],***

[1] Department of Aerospace Engineering, Sejong University, 209, Neungdong-Ro, Gwangjin-Gu, Seoul 05006, Korea; tuab@sejong.ac.kr
[2] Department of Aerospace Engineering, Pusan National University, Busan 46241, Korea; endale@pusan.ac.kr (B.E.); wondebly@pusan.ac.kr (A.W.)
[3] Department of Aerospace Engineering, and Convergence Engineering for Intelligence Drone, Sejong University, Seoul 05006, Korea
* Correspondence: hyhwang@sejong.edu; Tel.: +82-10-6575-2282

**Abstract:** The need for civilian use of Unmanned Aerial Vehicles (UAVs) has drastically increased in recent years. Their potential applications for civilian use include door-to-door package delivery, law enforcement, first aid, and emergency services in urban areas, which put the UAVs into obstacle collision risk. Therefore, UAVs are required to be equipped with sensors so as to acquire Artificial Intelligence (AI) to avoid potential risks during mission execution. The AI comes with intensive training of an on-board machine that is responsible to autonomously navigate the UAV. The training enables the UAV to develop humanoid perception of the environment it is to be navigating in. During the mission, this perception detects and localizes objects in the environment. It is based on this AI that this work proposes a real-time three-dimensional (3D) path planner that maneuvers the UAV towards destination through obstacle-free path. The proposed path planner has a heuristic sense of $A^\star$ algorithm, but requires no frontier nodes to be stored in a memory unlike $A^\star$. The planner relies on relative locations of detected objects (obstacles) and determines collision-free paths. This path planner is light-weight and hence a fast guidance method for real-time purposes. Its performance efficiency is proved through rigorous Software-In-The-Loop (SITL) simulations in constrained-environment and preliminary real flight tests.

**Keywords:** vision-based navigation; cluttered environment; three-dimensional path planner; obstacle avoidance; machine learning

## 1. Introduction

The cost-effectiveness, ease of access, and mission versatility are the primary compelling qualities of UAVs that attract many aerospace and related sectors. Hence, UAVs are being integrated into tasks such as package delivery, first aid, law enforcement, disaster management, infrastructure inspection, agriculture mechanization, rescue, military intelligence, and many more. As low-altitude aerial vehicles, however, UAVs often encounter obstacles such as trees, mountains, high storey buildings, electric poles, and so on during their missions. Therefore, these aerial vehicles should be equipped with sensors to perceive the environment around them and avoid potential dangers.

To leverage the use of UAVs in cluttered environments, studies have been conducted on the types and ways of integrating various sensors for autonomous navigation. Vehicle localization is one of the pillars of autonomous navigation. In an open-air space, Global Positioning System (GPS) is often used for UAV localization. However, GPS-based UAV localization in cluttered environment is unreliable. In such environment, sensors onboard the UAV are used for localization as well as collision avoidance. Ivan Konovalenko et al. [1] fused inputs from visual camera and Inertial Navigation System (INS) to localize a UAV. Based on computer simulation, the team analyzed various approaches to vision-based UAV

position estimation. Jinling Wang et al. [2] combined inputs from GPS, INS, and vision sensors to autonomously navigate UAVs. In their report, the inclusion of GPS input reduces vision-based UAV localization errors and hence enhances the accuracy of navigation. Jesus Garcia et al. [3] presented a methodology of assessing the performance of sensors fusion for autonomous flight of UAVs. Their methodology systematically analyzes the efficiency of input data for accurate navigation of UAVs.

Computer vision technology has evolved over the years to the stage that enables not only UAV localization but also obstacle detection and avoidance. This is realized through the advent of high-performance computers with the ability to process data and perform complex calculations at high speeds. With the promising progress in computer vision technology, many vision-based navigation algorithms have been developing. A comprehensive review of computer vision algorithms and their implementations for UAVs' autonomous navigation was presented by Abdulla Al-Kaff [4]. Lidia et al. [5] provided a detailed analysis on the implementation of computer vision technologies for navigation, control, tracking, and obstacle avoidance of UAVs. Wagoner et al. [6] also explored various computer vision algorithms and their capabilities to detect and track a moving object such as a UAV in flight.

Alongside computer vision technology, Artificial Intelligence (AI) is being implemented into UAVs navigation system to enable them to acquire humanoid perception. The idea is to train the computer that is either onboard a UAV or integrated with ground-based command system so that it takes control of UAV navigation with little to no human intervention. Su Yeon Choi and Dowan Cha [7] reviewed the historical development of AI and its implementation to UAVs with a particular focus on UAVs control strategies and object recognition for autonomous flight of UAVs. They also considered machine-learning-based UAV path planning and navigation methods.

The integration of AI and computer vision technology brings a remarkable importance in civilian application of UAVs. Many challenging tasks such as wildlife monitoring, disaster managment, and search and rescue are being addressed by UAVs equipped with AI and computer technology. Luis F. Gonzalez et al. [8] reported how AI- and computer-vision-enabled UAVs have solved the challenges of wildlife monitoring. The study reported by Christos and Theocharis [9] reflects the importance of UAVs equipped with AI and computer vision for autonomous monitoring of disaster-stricken areas. Eleftherios et al. [10] combined AI with a computer vision system onboard a UAV to enable real-time human detection during search and rescue operations.

The integration of the two aforementioned key technologies—AI and computer vision—provides environment acquaintance to UAVs. This helps the UAVs to plan their collision-free paths. For autonomous navigation, a UAV has to have either a predetermined path or a capacity to plan a path in real-time. A mission with predetermined route requires less number of sensors as compared to a mission with real-time path planning. The challenges with real-time path planning are the complexity of multiple sensors integration, input data synchronization, and computational burdens thereof. Valenti et al. [11] developed techniques to enrich a UAV with capabilities of localizing itself and autonomously navigate in a GPS-denied environment. In their report, stereo cameras on-board the UAV-based vision data were used for UAV localization and to build a 3D map of the surroundings. Based on this information, an improved $A^*$ path-planning algorithm was implemented for autonomous navigation of the UAV collision-free along the shortest path to the goal.

System-resource-intensive computational burdens on the companion computer onboard a UAV is always a setback to real-time path planning for the UAV. The companion computer has to deal with visual data processing for UAV localization, obstacle detection, and path planning. A comprehensive literature review on vision-based UAV localization, obstacle avoidance, and path planning was reported by Yuncheng et al. [12]. In their study, the challenges of acquiring real-time data processing for safe navigation of the UAV are reflected. They also reported the challenges of autonomous navigation of a UAV due

to intensive computation and high storage consumption of 3D map of the surroundings. Yan et al. [13] developed a computer-simulation-based deep reinforcement learning technique towards real-time path planning for UAV in dynamic environments. Although this is a promising step towards real-time path planning in dynamic environments, the assumption of predetermined global situational data and the absence of real flight test that verifies the efficiency the technique may degrade its attention.

To ease the computational burden on a companion computer dedicated to UAV localization, obstacle detection, and 3D path planning, we propose the integration of the fastest object detection algorithm with a light-weight 3D path planner that relies on few obstacle-free points to generate a 3D path. The proposed 3D path planner is based on AI acquired through YOLO (You Only Look Once ), which is the fastest object detection algorithm.

The study presented in this report is organized into sections. In Section 2, the problem to be addressed in this study is stated and the implemented methodology is explained. In Section 3, the overall descriptions of the implemented hardware and software components and their configurations are given. The machine learning approach for object detection is explained in Section 4. Then, the commonly known 3D path planning algorithms are discussed with their advantages and disadvantages in Section 5. The developed real-time 3D path planner is detailed in this section, followed by its performance tests in Section 6. Results and discussion are given in the final Section 7.

## 2. Problem Statement

The challenge in autonomous navigation of a UAV in urban environment is recognizing and localizing obstacles at the right time and continuously adjusting the path of the UAV in such a way that it can avoid the obstacles and navigate to the destination safely. To this end, it requires integrating effective object detection and path planning algorithms that run on a companion computer onboard the UAV.

Most of the widely used object detection algorithms are based on scanning the entire environment and discretizing the scanned region to create a dense mesh of grid points from which objects are detected. This process requires a companion computer with high storage capacity and intensive computational power. Moreover, the well-known path-planning algorithms either randomly sample or exhaustively explore the entire consecutive obstacle-free grid points to generate optimal path towards destination. This incurs additional computational burden on companion computer and compromises the real-timeness of the navigation commands . Liang et al. [14] conducted a comprehensive review on the most popular 3D path planing algorithms. In their review, a detailed analysis of the advantages and disadvantages of these commonly used algorithms is given. They reported that despite the intensive applications of these algorithms, the problem of real-time path planning in a cluttered environment remains unsolved.

Dai et al. [15] proposed light-weight CNN-based network structure for both object detection and safe autonomous navigation of a UAV in indoor/outdoor environments. However, the whole process of object detection and UAV path planning was performed on a ground-based computer and communication with the UAV was through a Wifi connection. This had a catastrophic drawback on the safe navigation of the UAV in indoor environment where Wifi connection failure is likely. Moreover, the Wifi data transfer rate may create a delay in navigation commands to be sent to the UAV. In an attempt to remove the dependency of the UAV on ground-based commands, Juan et al. [16] proposed a UAV framework for autonomous navigation in a cluttered indoor environment based on companion computer on-board the UAV. The performance of this framework was validated through hardware-in-the-loop simulation, and it appears to be promising to put an end to ground-based navigation command. However, an occupancy map of the cluttered environment in which the UAV navigated was pre-loaded on the companion computer. This undermines the applicability of the framework in dynamic environment.

To avoid computational burden on the companion computer, Antonio et al. [17] applied a data-driven approach, where data about the cluttered environment must be collected prior to the UAV mission. As proposed in their work, DroNet makes use of the collected data and safely navigates a UAV in the streets of a city. However, this approach, again, has limitations when it comes to dynamic or unknown environments.

This study, therefore, tends to address the challenges of computational burden subjected to companion computer onboard a UAV by integrating the available fastest object detection algorithm and the proposed light-weight real-time 3D path planner. Such an approach by-passes the challenges of dynamic or unknown environments. In the preliminary performance test, we assumed limited number of objects: pedestrian, window, electric poles, tunnel, trees, and barely visible nets as plausible obstacles that the UAV may encounter in a disaster monitoring scenario. Once the proposed 3D path planner is validated in a complete real-flight tests, further objects will be included in the machine learning process.

*Methodology*

To enable a companion computer onboard a UAV for simultaneous object detection and 3d path planning in real-time, it is essential to integrate the fastest object detection algorithm and 3D path planner that requires less computational burden. YOLO, as explained in Section 4.1, is selected as the fastest object detection algorithm. In addition to object detection, this algorithm also localizes the object(s). The proposed 3D path planner relies on the relative locations of the detected objects to calculate a collision-free path for the UAV. Although the proposed 3D path planner resembles $A^*$ path planning algorithm in implementing heuristic function for cost minimization, it avoids an exhaustive search for consecutive collision-free nodes and storage method of $A^*$. Unlike $A^*$, the proposed 3D path planner maps the current location of the UAV to a few nodes between consecutive obstacles. These few nodes are determined based on the size of the UAV and the gap between consecutive obstacles, as explained in Sections 5.1.1 and 5.1.2. A Euclidean function is used as a heuristic function in this 3D path planner.

Prior to a real flight test, the performance of the proposed 3D path planner must be checked in a simulated environment. For this performance test, software tools are essential components. One of the software tools specifically designed for such task is Gazebo 3D dynamic environment simulator. This software was primarily designed to evaluate algorithms for robots [18] and provides realistic rendering of the environment in which the robot navigates. Moreover, it is enriched by various types of simulated sensors. We designed a simulated cluttered 3D environment in Gazebo and used it to test performance of the proposed 3D path planner during its successive development.

## 3. Utilized Tools and Their Integration

Various open-source software was implemented in both the gazebo-based simulation and real flight tests for the development and validation of the 3D path planner. The type and implementations of this software are explained in the following two subsections.

### 3.1. Setup for Software-In-The-Loop Simulation

It is very common that Software-In-The-Loop (SITL) simulation is often used for testing the performance of an algorithm under development. This utility saves time and cost of repair of probable crashes in real flight test scenarios.

Open-source software such as px4 flight control firmware, Gazebo simulator, and Robot Operating System (ROS) were integrated and used for the development and performance testing of 3D path planner. Gazebo is a dynamic 3d model simulation environment particularly suitable for obstacle avoidance and computer vision. This simulation environment is enriched with simulated sensors that mimic the real sensors on-board the UAVs. YOLO object detector, with its Darknet architecture wrapped with ROS, was also used to publish information about obstacles in the UAV's navigation environment. For training

and validation of YOLO, images of the 3D models of the objects simulated in the gazebo simulation environment were taken. The Images were taken under various backgrounds and lighting conditions.

The 3D path planner algorithm that prompts px4 flight controller to send actuator commands to quadcopter model in gazebo simulator was developed as an ROS node. Hardware models implemented in this SITL Gazebo simulation are iris quadcopter, depth stereo camera, three ultrasonic sensors, and LiDAR as shown in Figure 1.

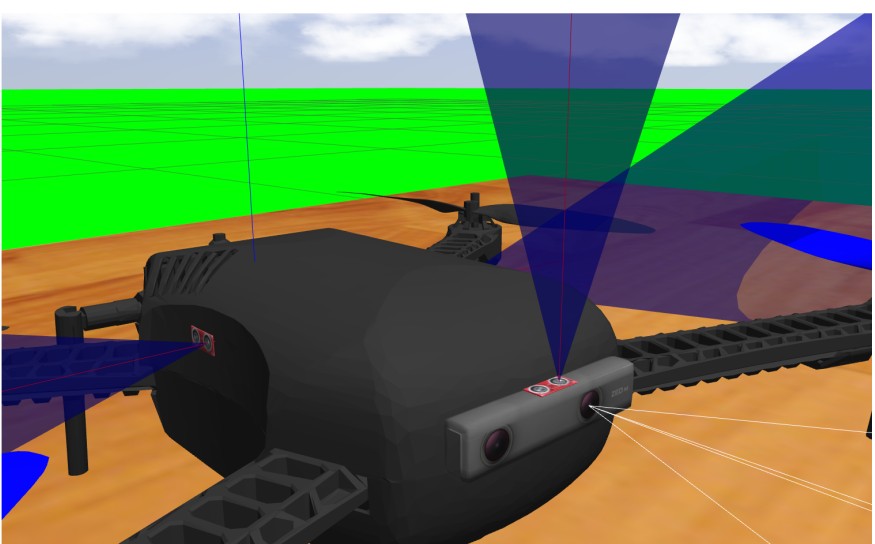

**Figure 1.** SITL: quadcopter equipped with on-board components.

The camera is for the frontal environment's image input, LiDAR is for quadcopter's altitude estimation in combination with GPS, and the ultrasonic sensors are used to detect lateral obstacles that may be encountered during takeoff and rolling. The 3D path planner acquires information from the aforementioned sensors in Gazebo simulator using Gazebo_ros packages that enables sensors to publish their information. The whole process runs on the desktop computer, whose software specifications are given in Table 1.

**Table 1.** Desktop Computer specification and software used for simulation.

| Type | Specification |
| --- | --- |
| Operating System | Ubuntu 16.04 |
| Memory | 2 GB |
| Processor | intel i7 CPU 972@2.67 GHz x 8 |
| Graphics | NV92 |
| Gazebo | version 7 with its dependencies |
| ROS | Kinetic with dependencies |
| PX4 firmware | version 1.9.2 |

### 3.2. Setup for Real-Flight Based Performance Test

Following SITL simulation-based performance validation, the 3D path planner was uploaded onto NVIDIA Xavier companion computer. The computer was integrated with Pixhawk 4 autopilot on-board Tarot 650 quadcopter platform. The platform components and their specifications are given in Table 2.

**Table 2.** Tarot quadcopter components specifications.

| Parameter/Item | Specification |
|---|---|
| Frame weight | 750 g |
| Motor to motor length | 600 mm |
| Payload weight | 1665.5 g |
| 4 motors | MN4006-23 KV: 380 T-motor |
| 4 propellers | 13 × 5.5 Carbon Prop |
| Battery | Poly-Tronics 14.8 V, 10,000 mAh |
| Electronic Speed Controller | Arris Simonk 30 A |

Hardware components used for real autonomous navigation are shown in Figure 2a. The Tarot quadcopter was equipped with a forward-looking ZED mini stereo camera, downward-looking LiDAR sensor, and upward-, right-, and left-looking ultrasonic sensors. The tasks of these hardware are as mentioned in the SITL simulation counterpart. The integration of the quadcopter, mounted sensors, and companion computer is shown in Figure 3.

The autopilot board is Pixhawk 4, which is mounted on the quadcopter underneath the companion computer: NVIDIA Xavier. The companion computer and LiDAR are connected to the Telem 2 and I2C ports, respectively, of autopilot board. ZED mini stereo camera and three ultrasonic sensors are connected to the companion computer.

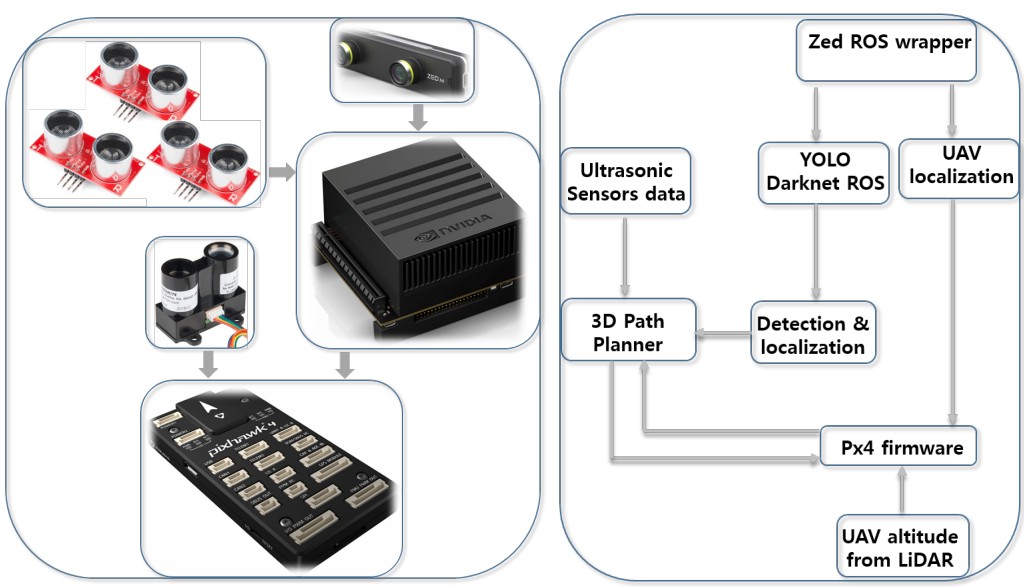

(**a**) Hardware architecture      (**b**) Software architecture

**Figure 2.** Architectures of hardware and software.

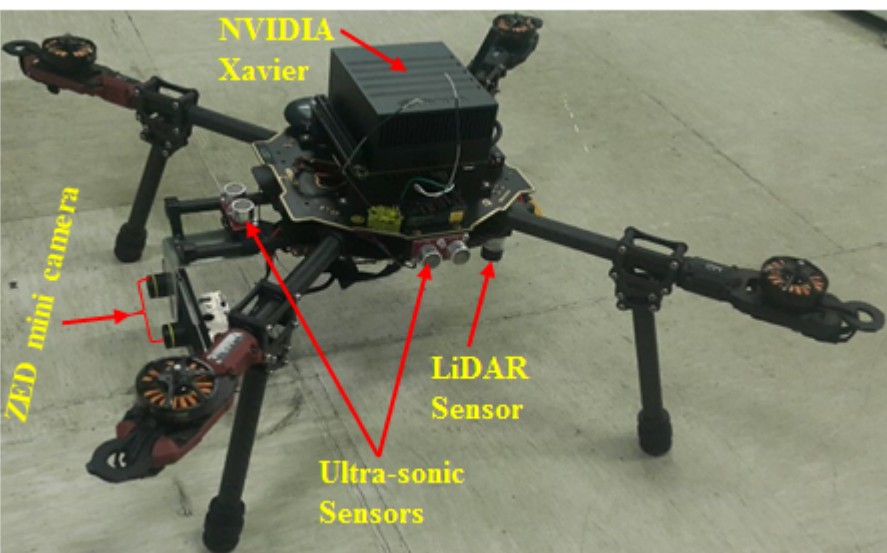

**Figure 3.** Quadcopter equipped with on-board components.

System Calibration and Configuration

PX4 firmware version 1.9.2 was installed on Pixhawk 4, and x-configuration type quadcopter airframe was selected. All the necessary sensors calibrations were done and parameters were set in such a away that the autopilot could communicate with external hardware. Quadcopter localization was enabled by GPS, LiDAR, and ZED mini stereo camera fusion. Pixhawk autopilot supports a Micro Aerial Vehicle Link (MAVLink) protocol that serializes messages. Telem 2 serial port of Pixhawk 4 was set to convey messages to-and-from Pixhawk through MAVLink protocol.

Robot Operating Software (ROS) Kinetic version was installed on the companion computer. ROS provides software tools that enable communication among hardware. Communication between the autopilot and companion computer was enabled by MAVROS: a ROS package that bridges ROS topics (message buses) with MAVLink messages. To extract information from the ZED mini stereo camera and publish in the form of specific message types through ROS topics, an open source, named ZED_ROS wrapper node, was installed on the companion computer. YOLO version 3 (YOLOv3) object detection algorithm and its framework Darknet_ROS were installed on the companion computer for obstacle detection and localization. The 3D path planner module runs on companion computer and communicates with autopilot through MAVROS. The configuration of software components is shown in Figure 2b.

## 4. Machine Learning for Object Detection

In the machine learning process, a companion computer onboard a quadcopter was trained to identify assumed obstacles that it may encounter during a disaster monitoring mission. The assumed obstacles are pedestrians, windows, electric poles, tunnels, trees, and barely visible nets. The companion computer can be trained to identify a large number of objects once the performance of the proposed path planner is validated on the assumed ones.

### 4.1. YOLO Object Detection and Localization

Object detection is a task in computer vision that involves identifying the presence and type of one or more objects in a given image. There are various types of object detection algorithms [19–25], and YOLO is one of them with its fastest detection and localization mechanisms. Matija Radovic et al. [26] reported the preference of YOLO over the other detection algorithms that runs on CNN. The key features underpinning YOLO as the fastest detection means are applying a single neural network on the entire image and

considering detected object localization as a regression problem. The architecture of this neural network is called Darknet: a type of CNN. It has 24 convolutional layers working as feature extractors and 2 dense layers for doing the predictions. A detailed discussion on the neural network and its architecture is given by Joseph Redmon et al. [27]. There is a configuration file with a given architecture. This file contains information about:

- layers and activations of the architecture
- anchor boxes
- number of classes
- learning rates
- optimization techniques
- input size
- probability score threshold
- batch size

Each configuration file has corresponding pre-trained weights. For training, YOLO requires two files: a file with list of names of objects and a file with a list of training images that contain desired objects with their corresponding labels. The labels are relative centers and dimensions of objects in the image. The configuration file can be modified as per the need of a user. For instance, increasing the batch value improves and speeds up the training but at the cost of demanding more memory. Two of the most important parameters in the configuration file that need to be checked are classes and final layer filters. The values of these parameters should match with the total number of objects in the training.

Once the training is over, the configuration and corresponding weight files are integrated with YOLO Darknet ROS module for object detection and localization during autonomous navigation of UAVs. Along with the detection of each object, there is a bounding box, which is characterized by the following parameters.

- confidence score that the object is detected
- center of the bounding box $(U_c, V_c)$
- dimension of the bounding box $(w, h)$

where $U$ and $V$ are coordinate axes of an image frame in which $U$ increases from left to right and $V$ increases from top to bottom. Both the center and dimensions of the box are normalized to fall between 0 and 1. Based on these parameters, the sides of the bounding box can be calculated as:

$$U_{min} = U_c - \frac{w}{2} \qquad \text{and} \qquad U_{max} = U_c + \frac{w}{2} \tag{1}$$

and

$$V_{min} = V_c - \frac{h}{2} \qquad \text{and} \qquad V_{max} = V_c + \frac{h}{2} \tag{2}$$

where $U_{min}$ and $U_{max}$ are the locations of left and right sides of the bounding box along the U-axis. Similarly, $V_{min}$ and $V_{max}$ are the locations of upper and lower sides of the bounding box along the V-axis. The coordinate transformation from the image frame to camera frame follows the procedure shown in [28]. Since the path planner was written as the ROS node that follows a reference frame FLU (Forward (x), left (y), and upward (z)), coordinate transformation from camera frame to ROS frame (FLU) was done. Moreover, PX4 uses FRD (Forward (x), right (y), and Down (z)). The ROS package MAVROS handles coordinate frame transformation from ROS frame to PX4 frame.

## 5. Three-Dimensional Path Planning Algorithms

The top challenge in autonomous navigation of UAVs is planning an obstacle-free route from the start to the destination. Encountering obstacles is possible, especially for missions like law enforcement, package delivery, and first aid in urban areas. Most of the path planning algorithms for UAVs are derived from pre-existing algorithms designed for ground robots. These algorithms are often 2D and need to be modified into 3D for aerial vehicles. The complexity to design and the demand for high performance computers

on-board the UAVs are challenges that incurred by the 3D path planners. The obstacle-free 3D path planning process demands an intensive computational burden that often limits the maximum cruising capability of the UAV. The effect of this computational burden is true for both free and cluttered environments as long as image processing has to occur.

Commonly known 3D path planning algorithms are $A^\star$ with its variants, Rapidly–Exploring Random Tree (RRT) with its variants, Probabilistic RoadMaps (PRM), Artificial Potential Field (APF), and Genetic or Evolutionary algorithms. These algorithms can be categorized into two: sampling-based and node/grid-base algorithms. Sampling-based algorithms connect randomly sampled points (subset of all points) all the way from start to the goal points thereby creating random graphs from which a graph with shortest path-length is selected. The algorithms include RRT, PRM, and APF.

Node/grid-based algorithms, unlike sampling-based algorithms, exhaustively explore throughout consecutive nodes. These algorithms include $A^\star$ and its variants. In search for an obstacle-free path, the algorithm takes in an image of the environment and discretizse it into grid cells that includes the current (start) location of the UAV and the goal location. The $A^\star$ algorithm has two functions to prioritize the cells to be visited. These two functions are the cost function, which calculates the distance from the current cell to the next cell, and the heuristic function, which calculates the distance from the next cell to the cell that contains the goal. With the objective of minimizing the sum of these two functions, the cells to be visited are heuristically prioritized. In the case of 3D search, the cost function calculates distances from the current cell to all 26 neighboring cells, and the heuristic function calculates distance from the 26 cells to the cell that contains the goal. In a cluttered environment with complex occlusion, highly dense grid cells are required, which in turn increase the computational burden, and thus the selected path may not be optimal.

*5.1. Machine Learning-Based 3D Path Planner*

Training an on-board computer to quickly identify objects and avoid collision with them in an environment in which UAV is set to navigate can be taken as a paradigm shift as it inherits the mechanism that a human being takes to avoid collision. The computational intelligence of a human brain is the degree that it is trained to, as is the artificial intelligence of the computer onboard a UAV. This is why intensive training of on-board computer is compulsory.

Apart from the capabilities of ensuring the presence of objects and their relative locations from the UAV, the companion computer may be required to know the type of objects it detected. The YOLO object detection algorithm installed on the companion computer has such a capability. Strategies to avoid collision with an object may depend on the type of the object. For instance, the avoidance mechanism for a window (open obstacle) is different from the mechanism for a tree (closed obstacle). Our 3D path planner includes those capabilities, as explained below.

5.1.1. Open Obstacles

In this type of obstacle, there is a possibility in which the UAV has no other option but to pass through the opening, such as in the case when the mission is to enter or exit a closed room through open window. Missions like in-house first aid or disaster monitoring may encounter such a scenario. In this case, the algorithm determines the relative position of the UAV with respect to the center of the bounding box around the obstacle. The center of the bounding box, as shown in Figure 4, has coordinate axes $(x_c, y_c, z_c)$ with respect to the ZED mini stereo camera frame, whose origin is located at the center of left camera.

The $x$, $y$, and $z$ axes of this frame point forward, right-to-left, and upward, respectively. Therefore, $x$ represents the depth of the detected object (e.g., $x_c$ depth of the window). The depth information is directly extracted from the ZED mini camera, whereas y and z are derived from the (U,V) coordinate values through coordinate transformation. Information obtained with respect to image frame, including Equations (1) and (2), are transformed to the camera frame.

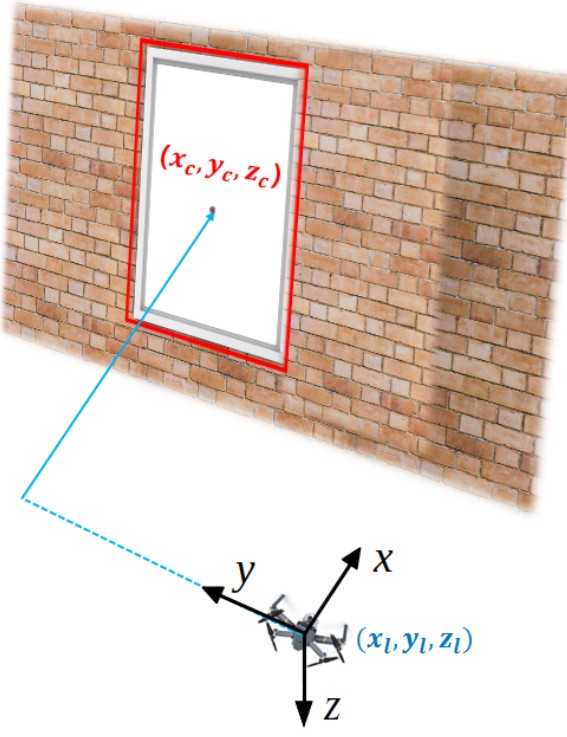

**Figure 4.** Open obstacle passing strategy.

The UAV's local position $(x_l, y_l, z_l)$ is acquired from GPS embedded in the Pixhawk 4 autopilot, LiDAR and ZED mini stereo camera. Before the UAV tries to pass through the window, it has to align itself with a vector normal to the plane of the window through appropriate attitude and altitude changes. In the figure, the setpoint $(x_l, y_s, z_s)$ is sent by the 3D path planner to the autopilot to command the UAV to adjust itself before advancing forward. The variables $y_s$ and $z_s$ are the y and z axes' setpoint values, respectively, obtained as follows:

$$y_s = y_l - y_c \qquad \text{and} \qquad z_s = z_l - z_c \tag{3}$$

While the UAV is responding to the command, the ultrasonic sensors mounted on the sides of the UAV check whether there are objects or not in the way. Once alignment is done, the UAV advances through the window with the setpoint $(x_s, y_s, z_s)$, where $x_s$ is the relative depth of the bounding box with clearance.

$$x_s = \mid x_l - x_c \mid + \quad obj_{clr} \tag{4}$$

The variable $obj_{clr}$ is a minimum object clearance or distance of the UAV behind the window that ensures the UAV has completely passed through the window with clearance. Moreover, to confirm the passage of the UAV through the window, the readings from the ultrasonic sensors mounted on the left and right sides of the UAV are considered. This method is implemented in cases like passing through tunnels or holes alike.

### 5.1.2. Closed Obstacles

If the obstacle is closed, our path planner considers the pass-by option with a minimum side clearance from the obstacle. The 3D path planning algorithm takes in bounding boxes information of all detected objects and assigns an identity index to each of them based on the locations of their centers along the y-axis. All information about the bounding box are with respect to the camera frame onboard the UAV. As shown in Figure 5a, the index value increases towards the increasing y-axis of the ZED min stereo camera (in this case, from right to left).

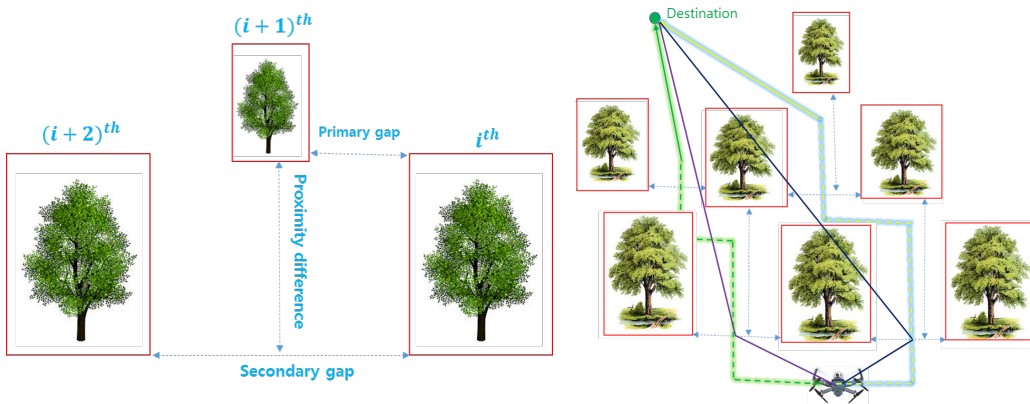

(**a**) Avoidance strategy

(**b**) Optimal path selection.

**Figure 5.** Obstacle-free optimal path selection

There are three conditions to be considered to determine next setpoint for the UAV. These are searching for

- Wide primary gaps: gaps between consecutive obstacles;
- Narrow primary gaps but with proximity difference: depth difference between consecutive obstacles; and
- Narrow primary gaps with small or no proximity difference.

Based on Figure 5a, the algorithm calculates the primary gap (between the $i^{th}$ and $(i+1)^{th}$) and secondary gap (between $i^{th}$ and $(i+2)^{th}$). The importance of calculating the secondary gap is that if the primary gap is narrow (less than twice UAV width) but with proximity difference more than twice the UAV length, there is the possibility that the UAV can advance forward but should check whether the secondary gap is wide enough or not to let the UAV pass in between. The gaps and proximity differences are calculated as follows:

$$
\begin{aligned}
Y_{min}^{i+1} - Y_{max}^{i} & \qquad \text{primary gap} \\
Y_{min}^{i+2} - Y_{max}^{i} & \qquad \text{secondary gap} \\
|\, X^{i+1} - X^{i} \,| & \qquad \text{proximity difference}
\end{aligned}
\tag{5}
$$

The pseudo-algorithm of our 3D path planner in the presence of multiple detected obstacles, as shown in Figure 5b, is given below.

- index the bounding boxes of the obstacles based on y-axis values of their centers. The box with the smallest y-axis value is indexed as the $i^{th}$ box;
- calculate $Y_{min}$ and $Y_{max}$ for each bounding box;
- calculate the primary gap between the $i^{th}$ and $(i+1)^{th}$
- if the gap is greater than or equal to twice UAV width;
  - calculate the midpoint of the gap;
  - calculate distances from the current location of the UAV to the midpoint and from the midpoint to the goal point. Save the sum of these two distances as path-length;
- else if the primary gap is smaller, calculate the proximity difference of the two consecutive bounding boxes $i^{th}$ and $(i+1)^{th}$;
  - if proximity difference is greater than or equal to twice UAV length, calculate the secondary gap;
  - if secondary gap is greater than or equal to twice the UAV width, check the following conditions:
    * if the $i^{th}$ obstacle is closer than the $(i+1)^{th}$, then set $(X^{i}, Y_{max}^{i} + obj_{clr}, Z^{i})$ as a potential setpoint;
    * else, set $(X^{i+2}, Y_{min}^{i+2} - obj_{clr}, Z^{i+2})$ as a potential setpoint;

- calculate distances from the current location of the UAV to the potential setpoint and from potential setpoint to the goal point. Save the sum of these two distances as path-length;
- apply the above steps for the remaining bounding boxes;
- compare the path-lengths and set the setpoint that leads to a minimum path length as the next setpoint for the UAV;
- else if the secondary gap is less than twice the UAV width, hover at a current altitude and yaw to search for any possible path applying the above procedure;
- if no path is discovered, land the UAV.

## 6. Path Planner Performance Tests

Performance tests were carried out during the developmental stage of the the path planner. Prior to real flight performance tests, rigorous computer-simulation-based tests were conducted. The implemented software tools and their integration as well as real flight test procedures are described in the following subsections.

### 6.1. SITL Test

Gazebo simulation environments shown in Figure 6 (front view) and Figure 7 (top views) were built, in which the path planner was to be tested.

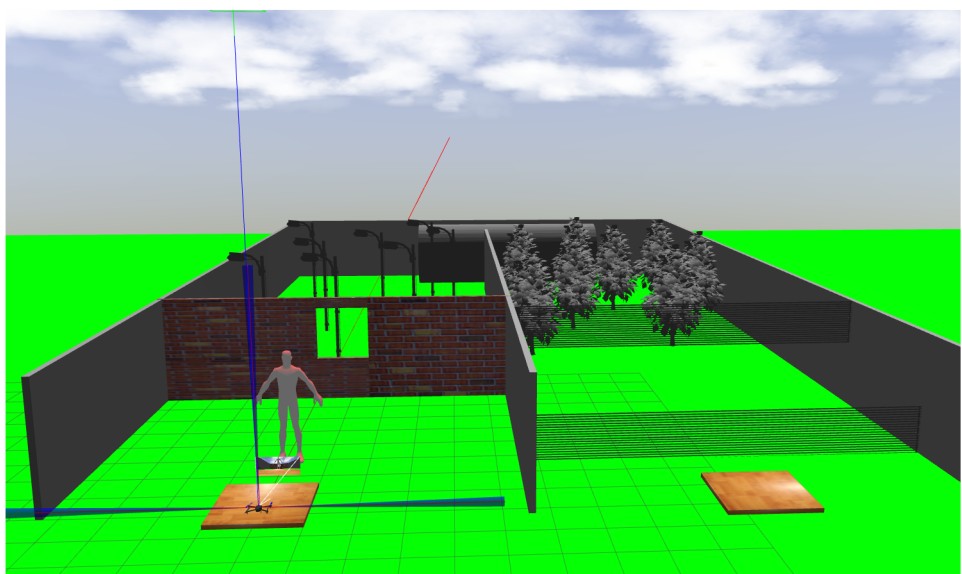

**Figure 6.** Front view of gazebo environment

The gazebo world has left and right sections. Each section has a width (*y*-axis) of 10 m and a length (*x*-axis) of 30 m. The UAV located in the left section has to avoid the obstacles on its mission to arrive at landing pad, which is located in the right section. During path planner performance tests, the poles and trees were randomly re-located in the simulation environment. Every time the arrangements of these obstacles are changed, the path followed by the UAV changes. Figure 7c,d shows two traced trajectories for the obstacles' arrangements shown in Figure 7a,b, respectively.

The 3D models of the obstacles imported to gazebo world were pedestrian, open window, poles, tunnel, trees, and two consecutive nets. The obstacles were designed in consideration of UAV mission for in-house first aid, law enforcement during suspect monitoring and door-to-door package delivery services in urban areas where the afore-mentioned obstacles are assumed to be potential threats in such missions. The UAV is supposed to pass by or through these obstacles on its way to a targeted location, in this case, the landing pad.

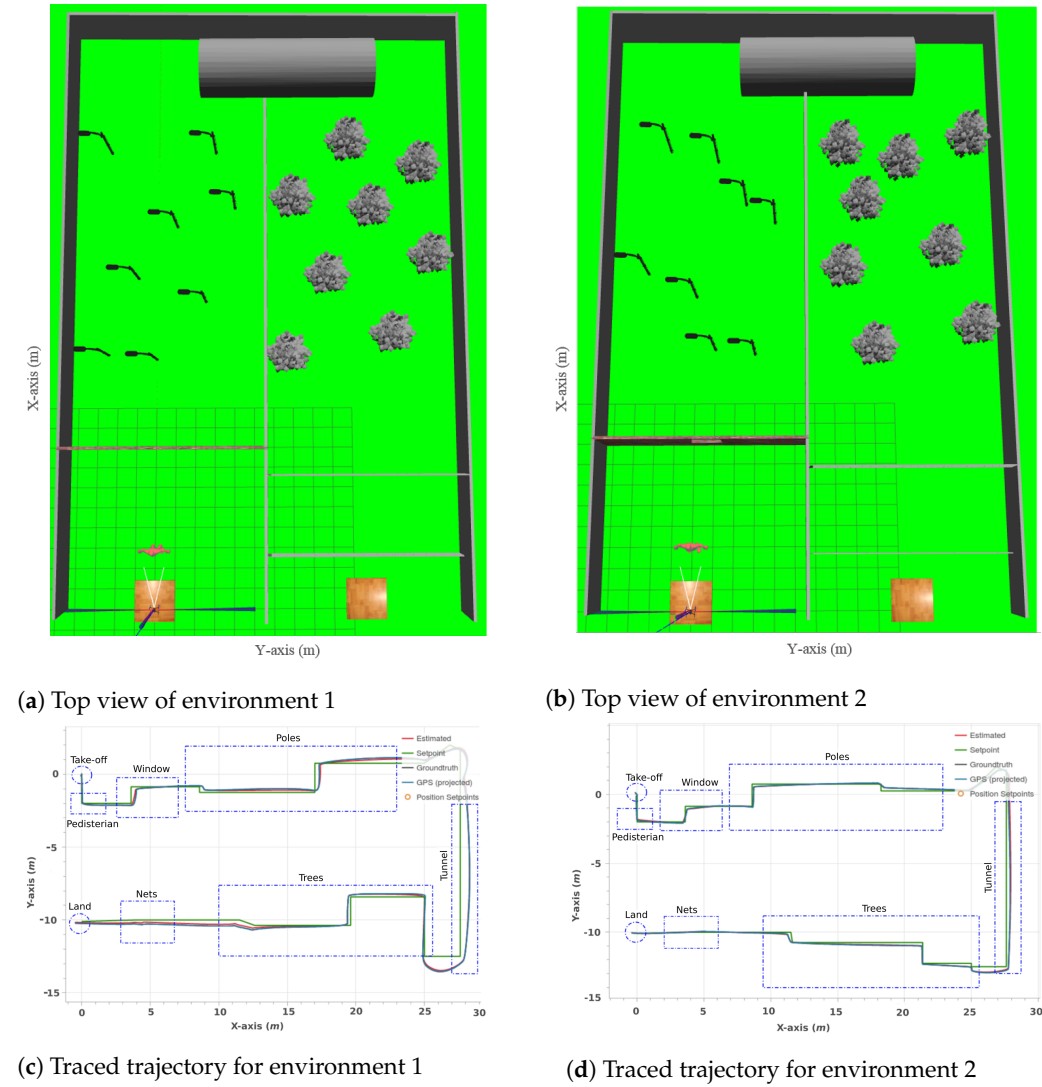

(**a**) Top view of environment 1

(**b**) Top view of environment 2

(**c**) Traced trajectory for environment 1

(**d**) Traced trajectory for environment 2

**Figure 7.** Top views of two simulation environments and traced trajectories.

The overall simulation infrastructure is shown in Figure 8. The 3D path planner written as ROS node communicates with the PX4 module named Mavlink_main. MAVROS bridges the ROS topics of the path planner with MAVLink messages of PX4 firmware. In addition to bridging ROS topics with MAVLink messages, MAVROS has extra-advantage in taking care of coordinate transformation between the ROS frame and PX4 Flight Control Unit (FCU) frame. ROS works with the East–North–Up (ENU) frame, and FCU works with the North–East–Down (NED) frame. PX4 firmware has a module called simulator_mavlink that lets the firmware interact with the 3D model of the UAV in the Gazebo world. The message exchanges between the PX4 firmware and gazebo simulator are handled by simulator MAVLink protocol.

As part of the 3D path planner's efficiency verification tests, video (named as Video S1) is submitted with this manuscript. The livestreamed videos on qgroundcontrol (a ground control station for UAVs) were recorded, and the snapshots of a video at the instants of attitude or altitude changes, to avoid obstacles, are displayed in Figure 9.

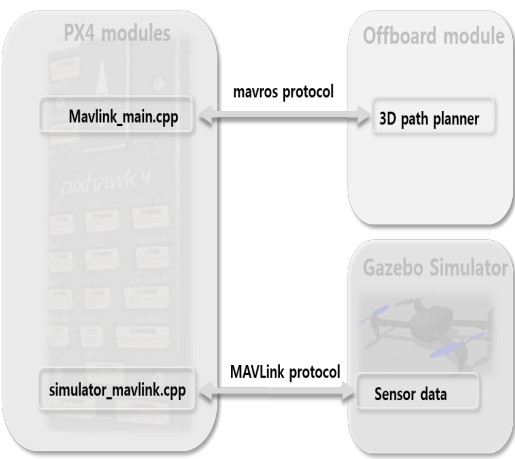

**Figure 8.** Software_In_The_Loop infrastructure.

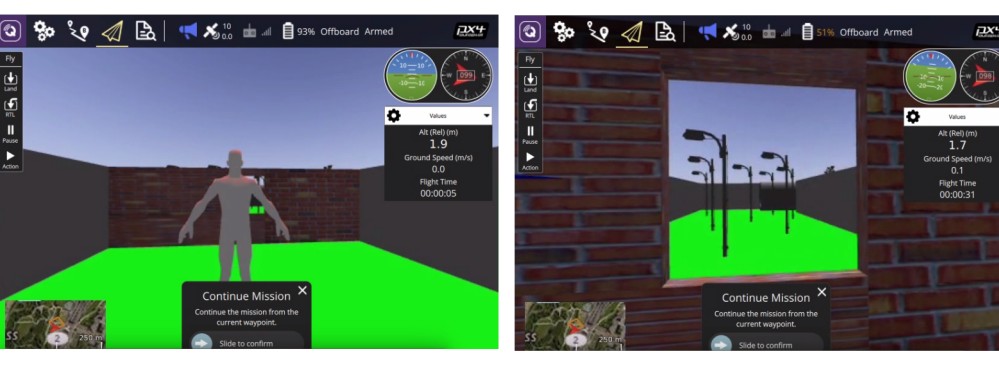

(**a**) Pass-by pedestrian

(**b**) Pass-through window

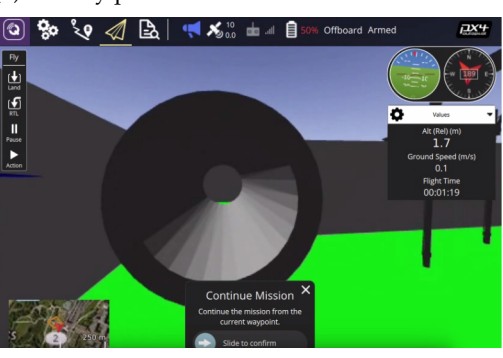
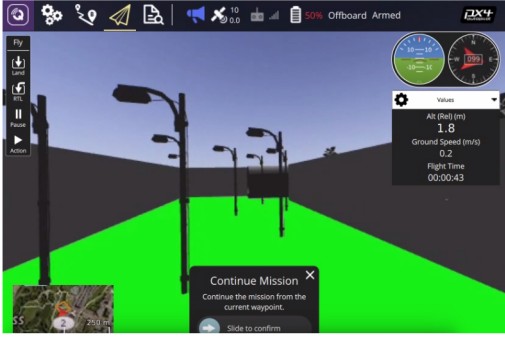

(**c**) Pass-through tunnel

(**d**) Pass-by poles

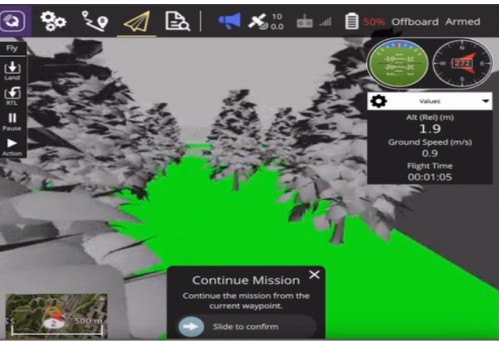
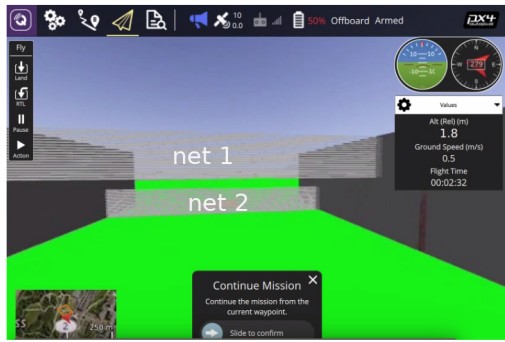

(**e**) Pass-through trees

(**f**) Net under/over pass

**Figure 9.** Instant snapshots during obstacle avoidance phases.

The position and attitude accuracy for the environments Figure 7c,d are shown in the first and second columns of Figure 10, respectively.

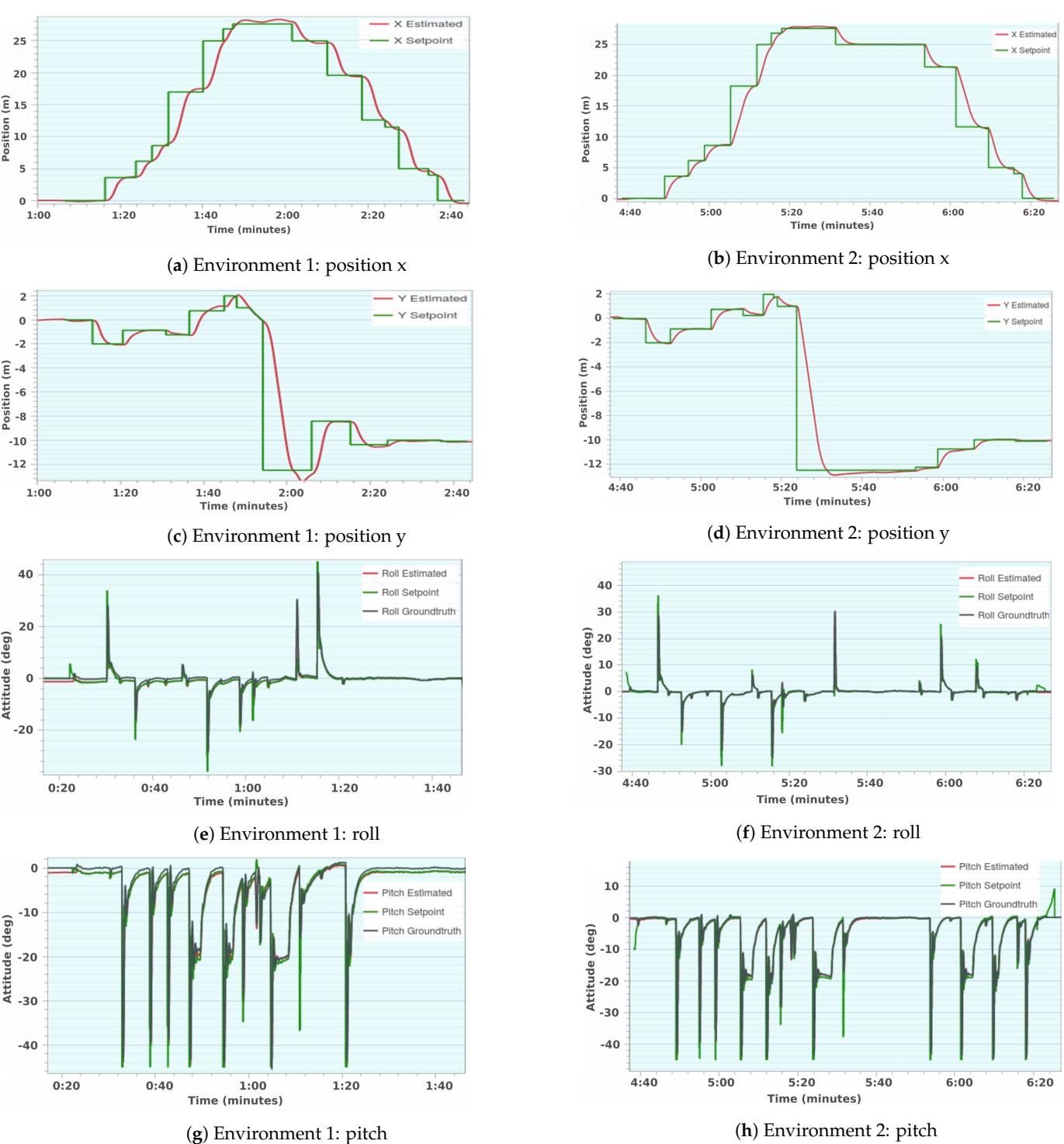

(**a**) Environment 1: position x

(**b**) Environment 2: position x

(**c**) Environment 1: position y

(**d**) Environment 2: position y

(**e**) Environment 1: roll

(**f**) Environment 2: roll

(**g**) Environment 1: pitch

(**h**) Environment 2: pitch

**Figure 10.** Simulation: position and attitude accuracy tests.

### 6.2. Real Flight Test

The real flight test requires us to do an intensive machine learning or training the companion computer to identify the obstacles simulated in the gazebo environment. This training process is not over yet: at least up to the report of this work. To get the sense of the efficiency of the path planner, real flight tests were conducted for the first obstacle pass,

as shown in Figure 11. As the quadcopter approaches the pedestrian, it has to evaluate the best route based on the conditions given in the pseudo-algorithm Section 5.1.2. For this test phase, short videos (named Videos S2 and S3) accompany this manuscript.

Considering the fact that building a real constrained environment as the simulated one requires time and money, a ROS node that sequentially publishes the simulated locations of obstacles was developed. The node publishes all the information that the 3D path planner requires from ZED mini stereo in a real scenario. Based on this, the quadcopter was deployed to arrive at a given destination, avoiding collisions with the obstacles. The effectiveness of the path planner is validated as shown in Figure 12, where the setpoints sent by the path planner and the estimated positions of the quadcopter throughout the whole mission overlap.

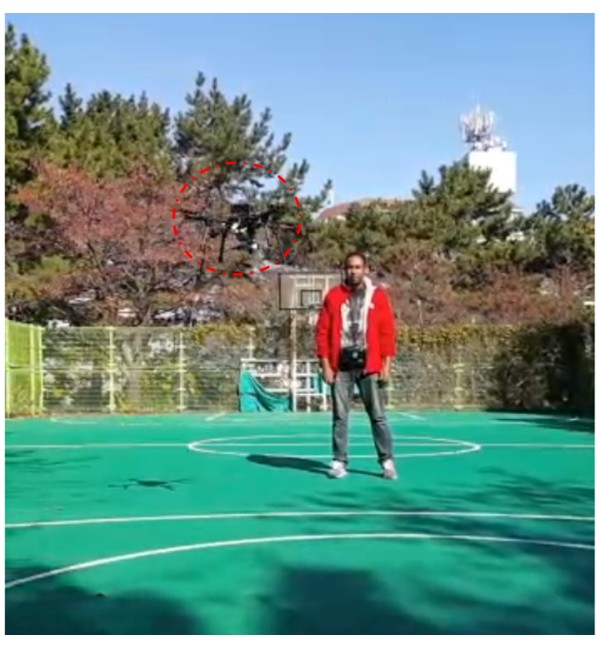
(**a**) Pedestrian pass test 1

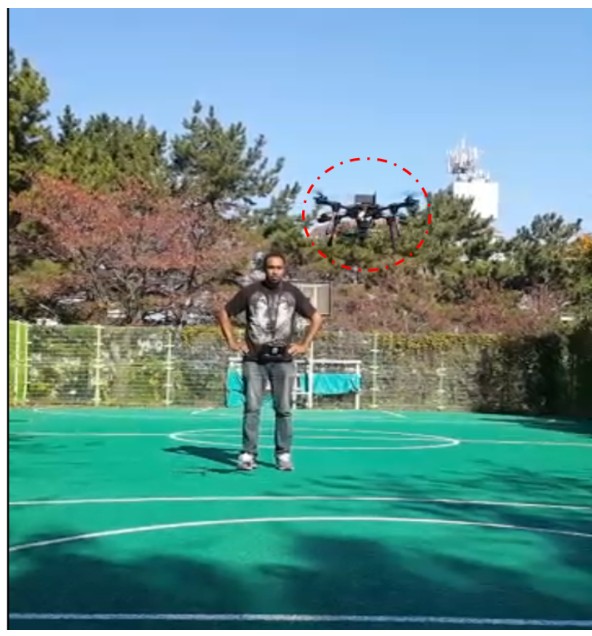
(**b**) Pedestrian pass test 2

**Figure 11.** Pedestrian as obstacle pass tests.

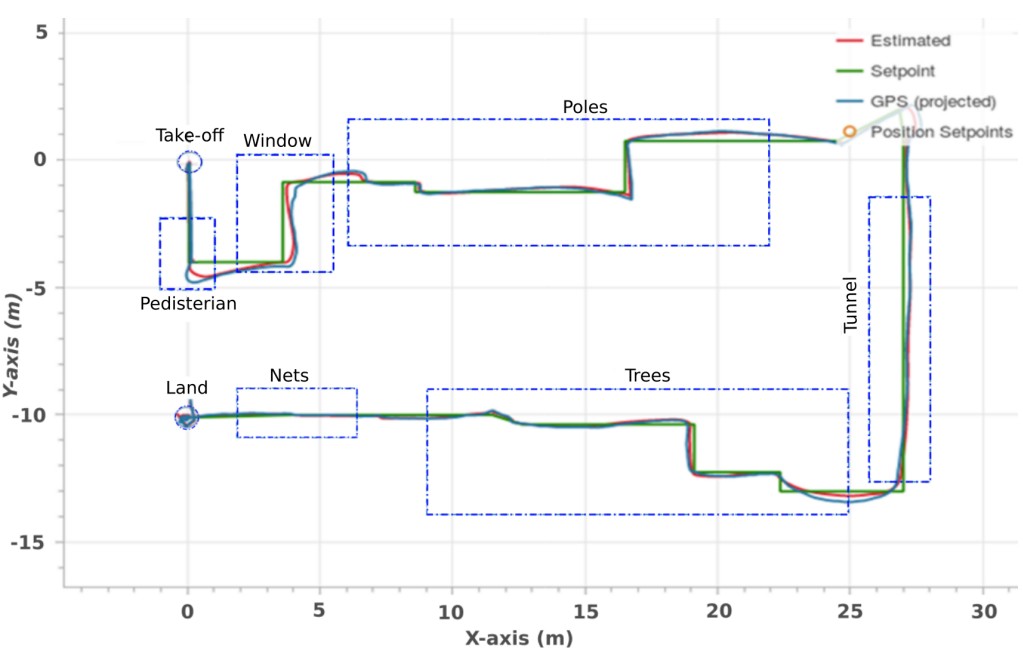

**Figure 12.** Estimated position and position setpoint comparison in real flight test.

Furthermore, Figure 13 shows component-wise position and attitude accuracy validation.

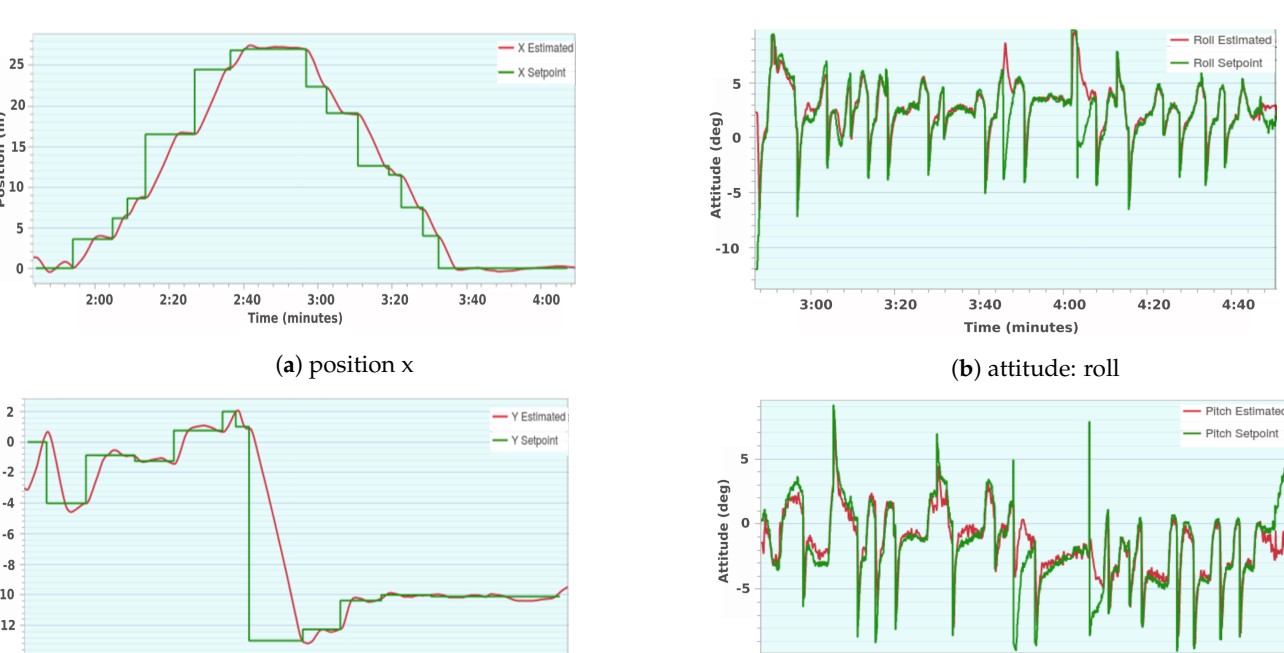

(**a**) position x

(**b**) attitude: roll

(**c**) position y

(**d**) attitude: pitch

**Figure 13.** Real flight test for path planner efficiency validation.

## 7. Results and Discussion

The validation of the developed 3D path planner was conducted through both SITL and preliminary real flight tests. Gazebo 3D model simulation environment was thoroughly used to develop and validate our 3D path planner prior to its upload into Pixhawk autopilot. The Gazebo environment shown in Figure 6 was set in such a way that it has obstacles like human, window, poles, tunnel, trees, and nets. These obstacles implicate the plausible encounters that the drone may face during missions such as package delivery, disaster monitoring, law enforcement, and first aid. For a complete navigation from the left section to the right section of the environment, the drone has to avoid collision with any of the mentioned obstacles and safely land at the landing pad.

Rigorous simulation tests were done where the two randomly arranged environments shown in Figure 7 are some of the environments in which the tests were done. The path followed by the quadcopter in the environment in Figure 7a is shown in Figure 7c. Similarly, the path followed in the environment in Figure 7b is shown in Figure 7d. As can be seen in these figures, the quadcopter followed two different trajectories in response to the two different arrangements of the obstacles in the environments. Moreover, the setpoints sent by the path planner and the estimated locations of the quadcopter overlap throughout the trajectories. This overlap validates the effectiveness of the path planner to autonomously navigate the quadcopter in a cluttered and GPS-denied environment.

Components of position and attitude responses in the two environments, Figure 7a,b, are shown in Figure 10. The well-traced setpoints of both position and attitude prove the efficiency of the path planner. In the path planner, a setpoint acceptance radius is set to 0.30 m. The differences observed at setpoint nodes are due to this acceptance radius. The quadcopter advances to the next setpoint assuming that the current setpoint is achieved at the moment the quadcopter crosses the acceptance radius, though the quadcopter may not reach the actual setpoint. This causes a gap between the estimated position and position setpoint. The attitude estimates of the quadcopter in both environments conform to the setpoints.

The preliminary real flight tests were conducted for collision avoidance with pedestrians. Machine learning was done for a pedestrian with different posture, clothes, and light exposures. As shown in Figure 9a, the quadcopter attempts to avoid collision with the pedestrian by rolling either right or left, implementing the conditions given in the pseudo-algorithm. For reference, the recorded two short videos on pedestrian collision avoidance are submitted with this manuscript.

For a complete mission test, a real environment, similar to the simulated environment shown in Figure 6, should have been constructed. This would take time and money. For this report, the real environment was modeled by an ROS node that publishes required information to the 3D path planner. This node publishes simulated locations of obstacles, and the 3D path planner takes those locations and calculates an obstacle-free path. With this, the UAV was commanded to autonomously head to the landing pad avoiding all possible obstacles on its way. The path followed by the UAV during this mission is shown in Figure 12. The overlap of the estimated quadcopter positions and intended setpoints shows that the 3D path planner effectively executed the mission.

In the real flight test, which was conducted in an open field, the quadcopter localization was limited to GPS and LiDAR. LiDAR is only for altitude estimation. ZED mini stereo camera, combined with GPS for quadcopter localization, does not provide proper localization of the quadcopter in an open field as it is required to get reflected rays from objects in its operation range. Therefore, for localization, the quadcopter in this circumstance relies on GPS whose accuracy is about 2 m. Depending on the number of satellites accessed and the environment in which the quadcopter is, the accuracy of the GPS drifts. The initial location of the quadcopter before takeoff had high drifts as can be seen in Figure 13c.

The test results obtained so far show that the 3D path planning algorithm is effectively guiding the UAV through collision-free paths. The future work includes the real flight tests in the environment similar to the simulated one as well as in unconstrained environments. Moreover, machine learning for various objects will be conducted based on the mission profile of the UAV.

**Supplementary Materials:** The following are available online at https://www.mdpi.com/article/10.3390/app11104706/s1, Video S1: Performance of path planner in cluttered environment, Video S2: Path planner in avoiding collision with pedestrian left pass, Video S3: Path planner in avoiding collision with pedestrian right pass.

**Author Contributions:** In this manuscript, machine learning for the aforementioned obstacles was done by B.E. and hardware integration and sensors fusion were done by A.W. The 3D path planning was done by A.T., while H.-Y.H. was responsible for the overall conceptual design of the methodology followed in this manuscript. All authors have read and agreed to the published version of the manuscript.

**Funding:** This work is supported by the Korea Agency for Infrastructure Technology Advancement (KAIA) grant funded by the Ministry of Land, Infrastructure and Transport (Grant 21CTAP-C157731-02).

**Institutional Review Board Statement:** Not applicable.

**Informed Consent Statement:** Not applicable.

**Data Availability Statement:** This study did not report any data.

**Acknowledgments:** This work is supported by the Korea Agency for Infrastructure Technology Advancement (KAIA) grant funded by the Ministry of Land, Infrastructure and Transport (Grant 21CTAP-C157731-02).

**Conflicts of Interest:** Authors mentioned in this manuscript have strongly involved in this study from the start to the end. The manuscript were thoroughly reviewed by the authors before its submission to journal of Applied Science. This manuscript has not been submitted to another journal for publication.

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
