# Peer review of "Machine Learning Approach to Real-Time 3D Path Planning for Autonomous Navigation of Unmanned Aerial Vehicle"

_applsci, doi:10.3390/app11104706_

Round 1
Reviewer 1 Report
“Machine Learning Approach to Real-time 3D Path Planning for Autonomous Navigation of Unmanned Aerial Vehicle”
The authors developed and demonstrated a system for active obstacle avoidance based on visual information.
Overall, the authors have put a serious effort towards the actual implementation/development of such a system. However, the theoretical contribution/analysis is absent. There is no formal “problem formulation” to provide the reader an exact representation of the problem to be tackled along with the corresponding assumptions. On the contrary, some theoretical analysis between the hardware/software components, e.g., lines 134-154, is really elementary and can be easily found in any machine learning/AI textbook.
Methodology:
What would happen if YOLO failed to detect an object (e.g., due to a “difficult” angle)? Probably for such tasks, a stereoscopic camera seems really important. What do you think? Could you utilize both approaches, incorporating some fusion mechanism?
Experimental validation:
My main concern is about the statistical significance of the results. I believe that the authors should first study the literature about this problem and provide the findings of this literature analysis in a separate section. After that, the authors should compare their methodology’s results along with some state-of-the-art approaches for validation purposes.
Author Response
Comments and Suggestions for Authors
“Machine Learning Approach to Real-time 3D Path Planning for Autonomous Navigation of Unmanned Aerial Vehicle”
The authors developed and demonstrated a system for active obstacle avoidance based on visual information.
Overall, the authors have put a serious effort towards the actual implementation/development of such a system. However, the theoretical contribution/analysis is absent. There is no formal “problem formulation” to provide the reader an exact representation of the problem to be tackled along with the corresponding assumptions. On the contrary, some theoretical analysis between the hardware/software components, e.g., lines 134-154, is really elementary and can be easily found in any machine learning/AI textbook.
Answer: Thank you so much for your fruitful comments. The problem statement is included into the revised manuscript as shown in lines 85 to 108 (paragraphs 8, 9, and 10) of introduction section.
As you commented, the statements given in lines 134 – 154 of the original manuscript can be easily found in textbooks. We briefly stated the overall concept of machine learning so that reads who have no theoretical background on machine learning will get the concept in just few lines and stay focused on the main work reported in the manuscript.
Methodology:
What would happen if YOLO failed to detect an object (e.g., due to a “difficult” angle)? Probably for such tasks, a stereoscopic camera seems really important. What do you think? Could you utilize both approaches, incorporating some fusion mechanism?
Answer: It is true that YOLO sometimes fails to detect objects, for instance when there is no enough luminosity or due to reflection. In such scenario, the developed path planner commands the quadcopter to roll either right or left based on information gained from two ultrasonic sensors mounted on the right and left sides of the drone. If detection fails as the quadcopter rolls left and right in the 10 m range of the width of the section of simulated environment, the quadcopter is commanded to quit the mission and land. Your suggestion is very promising . We will take the fusion mechanism into consideration in our future work as the present work is just a first phase of real-time path planning in cluttered environment.
Experimental validation:
My main concern is about the statistical significance of the results. I believe that the authors should first study the literature about this problem and provide the findings of this literature analysis in a separate section. After that, the authors should compare their methodology’s results along with some state-of-the-art approaches for validation purposes.
Answer: As you suggested, it would have been better to compare the obtained results with previous work. However, we could not find any related work based on YOLO object detection integrated with real-time 3D path planner.
Reviewer 2 Report
The authors present an interesting Machine Learning based 3D path planner for autonomous UAVs. The paper is interesting, well structured and globally well written, though it requires an attentive revision of the English language, since some typos were found (e.g.: "story building" should be "storey building", "qground" should be "ground", etc.).
The paper could be significantly improved if more information was given regarding the following aspects:
1) Athough most of the presented results are based on simulation, the authors have also built a prototype based on a Tarot quadcopter. What are the detailed characteristics of the prototype? What is the Tarot model (and relevant components, such as motor characteristics) used? What is weight of the payload to be carried? Were the physical constraints of the prototype UAV considered in the simulation model?
2) In the 3D path planner, which heuristic function was used to calculate the distances between the next cell and the cell that contains the goal?
Author Response
Comments and Suggestions for Authors
The authors present an interesting Machine Learning based 3D path planner for autonomous UAVs. The paper is interesting, well structured and globally well written, though it requires an attentive revision of the English language, since some typos were found (e.g.: "story building" should be "storey building", "qground" should be "ground", etc.).
Answer: Though, we were using American version, we replaced the word “story” by “storey” in the revised manuscript. The term “qground” is the name of ground control station for UAVs powered by either PX4 or Ardupilot flight controller. In the revised manuscript, however, explanation for the term “qgroundcontrol” is given in parentheses as shown in line 370.
The paper could be significantly improved if more information was given regarding the following aspects:
1) Athough most of the presented results are based on simulation, the authors have also built a prototype based on a Tarot quadcopter. What are the detailed characteristics of the prototype? What is the Tarot model (and relevant components, such as motor characteristics) used? What is weight of the payload to be carried? Were the physical constraints of the prototype UAV considered in the simulation model?
Answer: Thank you very much for your fruitful comments. Details about the Tarot quadcopter components are included in the revised manuscript as shown in Table 2. As for the physical constraints in the simulated environment, gazebo 3D model simulation environment has a simulated gravitational and quadcopter inertial effects. The weight, dimensions, inertial effects as well as collision characteristics of the quadcopter are incorporated to the Simulation Description Format(SDF) file of the modeled quadcopter in SITL_gazebo module of PX4 flight controller firmware. Moreover, SITL_gazebo has plugin libraries for motor and propeller specifications that can be customized to required values.
2) In the 3D path planner, which heuristic function was used to calculate the distances between the next cell and the cell that contains the goal?
Answer: Euclidean heuristic function was used. This is included in line 108 of the revised manuscript.
Reviewer 3 Report
This paper is about a novel system to mange autonomous navigation of unmanned aerial vehicle. This approach using machine learning to real-time 3D path planner to avoid the obstacle.
The article sounds interesting but lacks a few things. The reference literature is not detailed and does not show a good comparison with what already exists. It needs to be expanded and improved and compared with state-of-the-art methodologies.
It is not clear how innovative the methodology used is compared to what is present in the literature. It needs to be better explained and better motivated (why am I using neural networks? How do I apply them? What is my reference dataset?)
It is not clear what kind of experimental campaign was carried out. Add details.
Figures need to be improved in quality and presentation.
Author Response
Comments and Suggestions for Authors
This paper is about a novel system to mange autonomous navigation of unmanned aerial vehicle. This approach using machine learning to real-time 3D path planner to avoid the obstacle.
The article sounds interesting but lacks a few things. The reference literature is not detailed and does not show a good comparison with what already exists. It needs to be expanded and improved and compared with state-of-the-art methodologies.
Answer: Following your suggestion, the referenced literatures are presented in detail and in coherent way as shown in the revised manuscript . This is shown from the second to the tenth paragraph of the introduction section of the revised manuscript.
It is not clear how innovative the methodology used is compared to what is present in the literature. It needs to be better explained and better motivated (why am I using neural networks? How do I apply them? What is my reference dataset?)
Answer: The motivation behind this work and the approaches taken to address the problem stated therein are stated in the revised manuscript in lines 85 to 108 (paragraphs 8, 9, and 10) of introduction section.
It is not clear what kind of experimental campaign was carried out. Add details.
Answer: As stated in the manuscript, the conducted real-flight experiments were to check the performance of the developed path planner . Though, the experimental work is not completed yet, the test results obtained for pedestrian pass are appealing.
Figures need to be improved in quality and presentation.
Answer: We tried to enhance the quality and presentation of the Figures. However, we couldn’t see significant differences from the original ones.
Round 2
Reviewer 1 Report
- Again, the problem formulation is not clear. It would be better if you have a separate section that mentions the following: 1) Which are the assumptions that you make? 2) What is the problem that you try to solve? 3) Why this problem is difficult?
- I still believe that the elementary information should be kept out of the research paper.
- Regarding the experimental validation the authors stated the following: “As you suggested, it would have been better to compare the obtained results with previous work. However, we could not find any related work based on YOLO object detection integrated with real-time 3D path planner.” However, YOLO is part of your solution not the problem that you try to solve (that’s why you should have a clear-cut problem formulation, as I mentioned in the first bullet). Thus, the authors should compare their approach (YOLO-based real-time 3D planner) with other available UAV-obstacle-avoidance planners from the literature. Just to name a few:
- Dai, X., Mao, Y., Huang, T., Qin, N., Huang, D., & Li, Y. (2020). Automatic obstacle avoidance of quadrotor UAV via CNN-based learning. Neurocomputing, 402, 346-358.
- https://github.com/sarthak268/Obstacle_Avoidance_for_UAV
- Loquercio, A., Maqueda, A. I., Del-Blanco, C. R., & Scaramuzza, D. (2018). Dronet: Learning to fly by driving. IEEE Robotics and Automation Letters, 3(2), 1088-1095.
- Sasongko, R. A., Rawikara, S. S., & Tampubolon, H. J. (2017). UAV obstacle avoidance algorithm based on ellipsoid geometry. Journal of Intelligent & Robotic Systems, 88(2), 567-581.
Author Response
Comments and Suggestions
- Again, the problem formulation is not clear. It would be better if you have a separate section that mentions the following: 1) Which are the assumptions that you make? 2) What is the problem that you try to solve? 3) Why this problem is difficult?
Answer: In the revised manuscript, problem statement and approaches to address the problem are given in a separate section (section 2). We believe that this section encloses, the assumptions we made, the problem we intend to solve, and the challenges of the addressed problem.
Comments and Suggestions
- I still believe that the elementary information should be kept out of the research paper.
Answer: we removed the elementary information about the machine learning in the revised manuscript.
Comments and Suggestions
- Regarding the experimental validation the authors stated the following: “As you suggested, it would have been better to compare the obtained results with previous work. However, we could not find any related work based on YOLO object detection integrated with real-time 3D path planner.” However, YOLO is part of your solution not the problem that you try to solve (that’s why you should have a clear-cut problem formulation, as I mentioned in the first bullet). Thus, the authors should compare their approach (YOLO-based real-time 3D planner) with other available UAV-obstacle-avoidance planners from the literature. Just to name a few:
- Dai, X., Mao, Y., Huang, T., Qin, N., Huang, D., & Li, Y. (2020). Automatic obstacle avoidance of quadrotor UAV via CNN-based learning. Neurocomputing, 402, 346-358.
- https://github.com/sarthak268/Obstacle_Avoidance_for_UAV
- Loquercio, A., Maqueda, A. I., Del-Blanco, C. R., & Scaramuzza, D. (2018). Dronet: Learning to fly by driving. IEEE Robotics and Automation Letters, 3(2), 1088-1095.
- Sasongko, R. A., Rawikara, S. S., & Tampubolon, H. J. (2017). UAV obstacle avoidance algorithm based on ellipsoid geometry. Journal of Intelligent & Robotic Systems, 88(2), 567-581.
Answer: Following your suggestion, we referenced related research works and discussed their limitations as compared to our work. These are given in the section two of the revised manuscript.
Reviewer 3 Report
There are no significant changes from what has been suggested
Author Response
Comments and Suggestions for Authors
There are no significant changes from what has been suggested
Answer: In the revised version of the manuscript, relevant and related previous works are referenced and their advantages and limitations are discussed. Problem statement is included as a new section (section 2) in the revised manuscript. We hope that this will enable readers to understand the intention and designed approach of our research work clearly. As for the figures’ quality, we believe that the labels and annotations are legible.
Round 3
Reviewer 3 Report
The authors modified what was previously requested. I would have opted for an overall improvement in the images and overall presentation which lacks clarity in places overall is now better than before.
Try to improve the quality of the paper if possible
Author Response
Answer: As you stated, we made an overall improvement on the manuscript. Following your suggestion, we improved the quality of figures in the revised manuscript. As a result, the qualities of Fig. 7(a) and (b), Fig. 12, and Fig. 13 are improved. There were overlaps between subfigures in Fig.13. The gaps between the subfigures are now clearly visible in the revised manuscript.
As for the rest, we feel that the cited references are relevant to the paper and they are presented coherently. The problem statement on the separate section clearly reflects the motive and methodologies of this research work.